# Notch3 Transactivates Glycogen Synthase Kinase-3-Beta and Inhibits Epithelial-to-Mesenchymal Transition in Breast Cancer Cells

**DOI:** 10.3390/cells11182872

**Published:** 2022-09-14

**Authors:** Weiling Chen, Yongqu Zhang, Ronghui Li, Wenhe Huang, Xiaolong Wei, De Zeng, Yuanke Liang, Yunzhu Zeng, Min Chen, Lixin Zhang, Wenliang Gao, Yuanyuan Zhu, Yaochen Li, Guojun Zhang

**Affiliations:** 1Department of Breast-Thyroid-Surgery and Cancer Center, Xiang’an Hospital of Xiamen University, School of Medicine, Xiamen University, Xiamen 361101, China; 2Key Laboratory for Endocrine-Related Cancer Precision Medicine of Xiamen, No. 2000 Xiang’an East Road, Xiamen 361101, China; 3Xiamen Research Center of Clinical Medicine in Breast & Thyroid Cancers, No. 2000 Xiang’an East Road, Xiamen 361101, China; 4Department of Medical Oncology, Xiang’an Hospital of Xiamen University, No. 2000 Xiang’an East Road, Xiamen 361101, China; 5Department of Pathology, Cancer Hospital of Shantou University Medical College, No. 7 Raoping Road, Shantou 515041, China; 6Department of Medical Oncology, Cancer Hospital of Shantou University Medical College, No. 7 Raoping Road, Shantou 515041, China; 7Department of Breast and Thyroid Surgery, The First Affiliated Hospital of Shantou University Medical College, No. 57 Changping Road, Shantou 515041, China; 8Department of Central Lab, Cancer Hospital of Shantou University Medical College, No. 7 Raoping Road, Shantou 515041, China

**Keywords:** breast cancer, Notch3, GSK3β, epithelial-to-mesenchymal transition, prognosis

## Abstract

As a critical transformational process in the attributes of epithelial cells, epithelial-to-mesenchymal transition (EMT) is involved in tumor invasion, metastasis, and resistance to treatment, which contributes to the ultimate death of some patients with breast cancer. Glycogen synthase kinase-3-beta (GSK3β) is thought to be an EMT suppressor that down-regulates the protein, snail, a zinc finger transcription inhibitor, and regulates E-cadherin expression and the Wnt signaling pathway. Our previous studies have shown that Notch3 also inhibits EMT in breast cancer. In mammary gland cells, GSK3β physically bound and phosphorylated the intracellular domain of two Notch paralogs: N1ICD was positively regulated, but N2ICD was negatively regulated; however, the relationship between Notch3, GSK3β, and EMT in breast cancer is still unclear and crosstalk between Notch3 and GSK3β has not been widely investigated. In this study, we revealed that Notch3 was an essential antagonist of EMT in breast cancer cells by transcriptionally upregulating GSK3β. In breast cancer, MCF-7 and MDA-MB-231 cell lines, the silencing of Notch3 reduced GSK3β expression, which is sufficient to induce EMT. Conversely, ectopic Notch3 expression re-activated GSK3β and E-cadherin. Mechanistically, Notch3 can bind to the *GSK3β* promoter directly and activate GSK3β transcription. In human breast cancer samples, Notch3 expression is positively associated with GSK3β (*r* = 0.416, *p* = 0.001); moreover, high expressions of Notch3 and GSK3β mRNA are correlated to better relapse-free survival in all breast cancer patients via analysis in “the Kaplan–Meier plotter” database. In summary, our preliminary results suggested that Notch3 might inhibit EMT by trans-activating GSK3β in breast cancer cells. The suppression of Notch3 expression may contribute to EMT by transcriptionally downregulating GSK3β in breast cancer.

## 1. Introduction

Breast cancer is the most common malignant tumor and the main cause of death due to cancer worldwide in women; moreover, local and/or distant metastasis is the leading cause of death in patients with this disease [1]. In recent decades, advances in early diagnosis, improved surgical techniques, and better adjuvant drug therapy have decreased the mortality rate of breast cancer [2]. Unfortunately, many patients exist who still have relapses or distant metastases, resulting in eventual treatment failure. The process of epithelial-to-mesenchymal transition (EMT) results in the transformation of epithelial cells into a mesenchymal phenotype, significantly reduces adhesion strength between cells, and drives the aggressiveness and cell migration of epithelial tumors [3]. In recent years, EMT has been associated with tumor genesis and progression, and is considered to be an important biological process leading to epithelial malignancy during embryonic development, with enhanced migration and invasion capacity [4]. The mechanisms involved in the occurrence and progression of EMT have been extensively explored, but other potential mechanisms, especially critical genes that drive EMT, still remain unclear [5]. As a highly conserved signaling pathway, Notch plays an essential role in numerous biological processes in metazoan development, including stem cell self-renewal, cell differentiation, proliferation, migration, adherence, survival, apoptosis, and cell-fate decisions [6,7]. Recently, increasing evidence has shown that Notch plays a fundamental role in the development and tumorigenesis of the mammary gland, including induction of the EMT process [8,9]. In the canonical Notch signaling pathway, ligand binding induces the release and nuclear translocation of Notch receptor intracellular domains (NICDs), which interact with the transcription factor CSL, leading to subsequent activation of canonical Notch target genes. Liu et al. confirmed the finding that Notch1, Notch2, and CSL usually bind to different target genes, suggesting that the signaling activity of most Notch family members is relatively independent [10]. Similarly, genome-wide identification of Notch transcriptional complex binding sequence pairing sites has shown that Notch family members have different binding sequence pairing sites [11,12]. Notch3, a member of the Notch family of transmembrane receptors, is considered an oncogene or tumor suppressor in distinct cancers [13,14,15,16]. Our previous studies showed that estrogen receptor (ER)α inhibits EMT by suppressing Bmi1 [17], and Notch3 transcriptionally upregulates ERα in breast cancer. Furthermore, Notch3 can inhibit EMT in breast cancer epithelial cells by transactivating GATA3 [18]. The aforementioned results indicate that Notch3 inhibits the progression of EMT in breast cancer.

The key factor of the Wnt signaling pathway, glycogen synthase kinase-3-beta (GSK3β), is a serine/threonine kinase that can phosphorylate vast numbers of substrates, such as structural proteins, transcription factors, and signaling proteins [19]. Growing evidence demonstrates that GSK3β acts as an EMT suppressor via its downregulation of snail, a zinc-finger transcriptional repressor that regulates the epithelial marker, E-cadherin [20]. Both Notch and Wnt pathways are known to play key roles in development and disease [6]. Direct or indirect interactions have been shown between these two signaling pathways [21,22]; however, the relationship between Notch3 and GSK3β in EMT of breast cancer is still unclear.

The purpose of this study was to investigate whether Notch3 acts as a transcriptional activator of GSK3β in breast cancer. Notch3 was found to inhibit EMT by directly binding to the *GSK3β* promoter and transactivating GSK3β in breast cancer. In addition, Notch3 and GSK3β expression was positively associated in human breast cancer samples. Finally, over-expression of both Notch3 and GSK3β mRNA was found to be associated with better recurrence-free survival (RFS) in all patients with breast cancer.

## 2. Materials and Methods

### 2.1. Cell Lines, Plasmids, and Reagents

Human breast cancer cell lines MCF-7, MDA-MB-231 were purchased from the American Type Culture Collection (Manassas, VI, USA). The pCMV and pCMV-Notch3 intracellular domain (N3ICD) plasmids were given by Professor Michael M. Wang from the Medical School of Michigan University. The pCLE-N3ICD (Plasmid 26894) and control vector pCLE (Plasmid 17703) were purchased from Addgene (Cambridge, MA, USA). The psi-U6.1/eGFP/short hairpin (sh)RNA-GSK3β plasmid was purchased from GeneCopoeia (Guangzhou, China), and pCMV3-GSK3β-GFPSpark was purchased from Sino Biological Inc (Beijing, China). A wild-type reporter assay vector containing a *GSK3β* promoter region from −353 to −268 (pGL3-GSK3β-enhancer) was generated with a DNA-binding protein (CSL) antisense binding sequence, TTCCCA. All antibodies used in this study are summarized in Appendix A.

### 2.2. Plasmid DNA and siRNA Transfection

We transfected 5 μg of pCMV or pCMV-N3ICD into MDA-MB-231 cells using Lipofectamine 2000 or Lipofectamine 3000 (Invitrogen, Waltham, MA, USA). Stable Notch3-overexpressing clones were established using DMEM plus 10% fetal bovine serum (FBS) containing 1 mg/mL (selected concentration) and 0.5 mg/mL (established concentration) of G418 (Merck&Co, Whitehouse Station, NJ, USA). In a rescue experiment, 2.5 μg of psi-U6.1/eGFP/shRNA-GSK3β was transfected into stable Notch3-overexpressing MDA-MB-231 cells growing in a 6-well cell culture plate for 48 h using Lipofectamine 3000 (Invitrogen). Control small interfering (si)RNA and Notch3 small interfering (si)RNA were synthesized by Gene Pharma Biotech (Shanghai, China). The siRNA sequences are listed in Appendix A. Control or Notch3 siRNA (100 pmol) was transfected into 50–60% confluent MCF-7 cells. In an MCF-7 rescue experiment, 2 μg of pCMV3-GSK3β-GFPSpark and 80 pmol of Notch3 siRNA were co-transfected into MCF-7 cells cultured for 48 h.

### 2.3. Western Blot

Western blotting was performed as previously described [23]. The antibodies used are outlined in Appendix A.

### 2.4. RNA Extraction and RT–PCR Analysis

Total RNA was isolated from cultured cells using TRIzol reagent (Invitrogen, Carlsbad, CA, USA) according to the manufacturer’s instructions. A PrimeScript RT Reagent Kit (Takara, Dalian, China) was used to synthesize first-strand cDNA. Gene expression was detected by quantitative real-time PCR analysis using a SYBR Select Master Mix (Applied Biosystems, Foster City, CA, USA). Sequences of all primers used in the study are summarized in Appendix A.

### 2.5. Immunofluorescence

Immunofluorescence staining of MCF-7 and T-47D cells was performed as previously described [24]. Staining was visualized with a Zeiss microscope (Zeiss, Oberkochen, Germany). Notch3 and GSK3β antibodies are listed in Appendix A.

### 2.6. Wound Healing Assay

Wound healing assays were performed to evaluate the migration of MDA-MB-231 and MCF-7 cells under different transfection conditions. MDA-MB-231 cells were transfected with pCMV, pCMV-N3ICD, or pCMV-N3ICD plus psi-U6.1/eGFP/shRNA-GSK3β, respectively, and inoculated in a 6-well cell culture plate at a density of 1 × 10^5^ cells/well. Subsequently, cells were incubated in serum-free DMEM for 12 h. A pipette tip was used to establish a line in a monolayer of adherent cells. Photos were taken at 0, 24 h (MDA-MB-231) or 48 h (MCF-7). Similarly, MCF-7 cells were transfected with siRNA-negative control (NC), siRNA-Notch3, and siRNA-Notch3 plus pCMV3-GSK3β-GFPSpark. Experiments were performed in triplicate with three follow-up fields recorded for each well.

### 2.7. Migration and Invasion Assays

BD Falcon Cell Culture Inserts (24-well plates, 8 μm; BD Biocoat, BD Biosciences, Franklin Lakes, NJ, USA) were used for cell migration assays as described previously [25,26]. In an invasion assay, cells were inoculated in the upper compartment of Matrigel-coated inserts (24-well plates, 8 μm; BD Biocoat, BD Biosciences). All experiments were conducted in three separate assays.

### 2.8. Chromatin Immunoprecipitation

Chromatin immunoprecipitation (ChIP) assays were performed as previously described [27,28] Design and validation of ChIP primers are provided in Appendix A. Sequences of all primers used in this study are shown in Appendix A.

### 2.9. Construction of Reporter Assay Vector

A reporter assay vector, pGL3-GSK3β-enhancer, containing a CSL-binding site region of the *GSK3β* promoter was constructed to further explore the relationship between Notch3 and GSK3β by dual-luciferase reporter assay. The structure of the vector and validation of the amplified fragments are provided in Appendix A.

### 2.10. Dual-Luciferase Reporter Assays

To evaluate the activity of the *GSK3β* promoter activated by Notch3, we performed a dual-luciferase reporter assay. MDA-MB-231 and MCF-7 cells were inoculated into 24-well cell culture plates, and MDA-MB-231 cells were transiently transfected with pCMV, pCMV-N3ICD, pGL3-GSK3β-enhancer-promoter reporter, and Renilla luciferase reporter plasmids using Lipofectamine 2000. Similarly, MCF-7 cells were transfected with the siRNA-NC, siRNA-Notch3, pGL3-GSK3β-enhancer-promoter reporter, and Renilla luciferase reporter plasmids using Lipofectamine 2000. Cells were harvested 48 h after transfection and measured as previously described [18].

### 2.11. Immunohistochemistry

Human breast cancer specimens were obtained from 68 patients who underwent breast cancer surgery at the Cancer Hospital of Shantou University Medical College in 2014. Immunohistochemical methods were undertaken as previously described [29].

The expression of Notch3 and GSK3β was detected, with anti-mouse or rabbit IgG antibody (Abcam) used as negative controls. The immunohistochemical analysis criteria of Notch3 and GSK3β were previously described [30]. Both Notch3 and GSK3β stains are localized in the cell nucleus and cytoplasm. The mean percentages of positive cells were scored as 0 (≤5%), 1 (5–24%), 2 (25–49%), 3 (50–74%), or 4 (>75%). The staining intensity was scored as 1 (weak), 2 (moderate), or 3 (strong). Final histological (h) scores were obtained for each case by multiplying the percentage and intensity score. Protein expression levels were further analyzed by classifying h values as low (based on an h value <5) or high (based on an h value ≥5). The antibodies used are listed in Appendix A.

### 2.12. Database Analysis

The patients’ overall survival, relapse-free survival, and distal metastasis survival rate was based on Notch3 and GSK3β expression levels, and automatically drawn by the database website’s software for Kaplan-Meier plots from the Kaplan–Meier Plotter (http://kmplot.com; accessed on 1 April 2022). The entire analysis process was performed in strict accordance with the site’s instructions. The Breast Cancer Gene-Expression Miner v4.7 database (http://bcgenex.centregauducheau.fr/BC-GEM/GEM-requete.php; accessed on 1 April 2022) was used to explore the relationship, at the DNA level, between Notch 3 and GSK3β.

### 2.13. Statistical Analysis

Statistical analysis involved the use of SPSS 19.0 (SPSS Inc., Chicago, IL, USA). Differences between variables were assessed by χ2 analysis or two-tailed Student’s *t*-test. Data are presented as the mean ± standard error of the mean (SEM) unless otherwise indicated. Two-sided *p* < 0.05 was considered statistically significant. Each experiment was done at least in triplicate.

## 3. Results

### 3.1. Elevated Expression of Notch 3 and GSK3β Correlated with a Luminal Subtype in Breast Cancer Cell Lines

We first performed an extensive analysis of the Breast Cancer Gene-Expression Miner v4.7 database to explore the relationship between Notch 3 and GSK3β. We found that Notch3 was highly correlated with GSK3β at the DNA level (*r* = 0.15, *p* < 0.0001; Figure 1A,B). To further investigate the association of Notch3 and GSK3β in breast cancer, we evaluated the protein and mRNA expression levels of GSK3β in MCF-7 and MDA-MB-231 cell lines with high and low expression of Notch3. Interestingly, GSK3β expression paralleled Notch3 expression, with high levels observed in MCF-7 cells and nearly absent levels in MDA-MB-231 cells (Figure 1C,D). To further explore Notch3 and GSK3βexpression, we performed immunofluorescence staining and as shown in Figure 1E, Notch3 and GSK3β were co-expressed in luminal breast cancer cell lines MCF-7 and T-47D.

### 3.2. Ectopic Notch3 Induces GSK3β Expression and Inhibits EMT in Human Breast Cancer Cells

Increasing evidence shows that GSK3β is an EMT suppressor and has been found to regulate Notch1 and Notch2 expression. We investigated whether crosstalk existed between Notch3 and GSK3β in the progression of EMT in breast cancer. We found that the protein level of GSK3β increased in overexpressed-N3ICD MDA-MB-231 cells but decreased in Notch3-knockdown MCF-7 cells (Figure 2B,E). Similarly, quantitative real-time PCR showed that the GSK3β mRNA level was consistent with the protein level detected by a western blot (Figure 2A,D).

To explore the role of Notch3 in breast cancer and its effect on EMT progression in breast cancer cells, we silenced Notch3 in MCF-7 cells using siRNA. Suppression/inhibition of Notch3 by siRNA significantly resulted in E-cadherin down-regulation and vimentin up-regulation, as shown in the figure (Figure 2C). Conversely, overexpression of N3ICD in MDA-MB-231 cells by transfecting pCLE-N3ICD led to E-cadherin up-regulation but vimentin down-regulation (Figure 2F). A noteworthy finding demonstrated that ectopic overexpression of N3ICD resulted in an increase of β-catenin phosphorylation (Thr41/Ser37/Ser33), ultimately decreased total β-catenin, and conversely, silencing of Notch3 by siRNA resulted in decreased β-catenin phosphorylation (Thr41/Ser37/Ser33), ultimately increased total β-catenin expression (Figure 2C,F). Immunofluorescence confirmed that Notch3 and GSK3β co-expressed in MCF-7 and T-47D cells, whereas Notch3 silencing also resulted in decreased GSK3β expression (Figure 2G); these data further demonstrate that Notch3 may induce GSK3β expression. N3ICD overexpression downregulates the mesenchymal phenotype marker and upregulates the epithelial marker, which suggests that Notch3 may inhibit the progression of EMT in breast cancer.

### 3.3. Notch3 Activates GSK3β by Directly Binding to CSL-Binding Sites in the GSK3β Promoter

Based on our results that GSK3β expression is regulated by Notch3 transcriptionally, we speculated on whether Notch3 bound the *GSK3β* promoter. To search for a specific binding site in the *GSK3β* promoter we found a sense binding site at positions −1131 to −1126 and three antisense binding sites at −1165 to −1160, −308 to −303, and −167 to −162 (Figure 3A). To investigate whether Notch3 binds to the endogenous *GSK3β* promoter in chromosomal DNA, we performed a ChIP assay. We found that Notch3 interacts with the region between positions −368 to −268 containing a CSL antisense binding site (Figure 3B).

To investigate the activity of the *GSK3β* promoter directly regulated by Notch3, we constructed a reporter assay vector pGL3-GSK3β-enhancer containing the region from −354 to −268 of the *GSK3β* promoter. A dual-luciferase reporter assay was carried out to determine *GSK3β* promoter activity in Notch3-overexpressing MDA-MB-231 cells, and Notch3-knockdown MCF-7 cells by co-transfection with a reporter assay vector and Renilla luciferase reporter genes. After Notch3 silencing in MCF-7 cells, the luciferase activity of the GSK3β reporter decreased in a dose-dependent manner (Figure 3C). Consistently, the luciferase activity of the GSK3β reporter increased following N3ICD overexpression in MDA-MB-231 cells (Figure 3D); these results indicated that Notch3 upregulated GSK3β by directly binding to the *GSK3β* promoter.

### 3.4. Ectopic N3ICD Expression Inhibits Migration and Invasion In Vitro, Which Is Attenuated by GSK3β Silencing

To investigate the effects of N3ICD overexpression or knockdown of Notch3 on migration and invasion by breast cancer cells, a wound healing assay, transwell in vitro migration, and invasion assays were performed. After cells were cultured for 48 h, the width of the wound in MCF-7 siRNA-Notch3 cells had reduced to only 25% compared to that at 0 h, while the width of the wound in siRNA-NC cells had reduced to about 54.5% compared to that at 0 h; this suggested that silencing of Notch3 disinhibited MCF-7 cells that had migrated into the wound area. Conversely, the effect was reversed by ectopic GSK3β expression as the width of the wound was restored to 43.5%, indicating that the increased migration caused by the silencing of Notch3 was partly compensated by GSK3β (Figure 4A,B). In contrast, we observed that after culturing for 24 h, overexpression of N3ICD inhibited MDA-MB-231 cell migration into the wound region compared with that of the control group (44% vs. 14.5%), while downregulation of GSK3β expression reversed this (21.5%; Figure 4E,F). We initially evaluated the effects of recombined Notch3 and GSK3β expression on breast cancer cell invasion using migration and invasion assays. Notch3 knockdown increased cell migration and invasion by 3.87-fold and 5.35-fold, respectively, which was partially eliminated by GSK3β overexpression (Figure 4C,D). In contrast, N3ICD overexpression reduced migration and invasion by nearly 80.9% and 78.43%, respectively, which was partially abrogated by GSK3β silencing (Figure 4G,H); these data demonstrated that Notch3 induces GSK3β expression and inhibits wound healing and migration/invasion, an effect that can be attenuated by GSK3β knockdown in vitro.

### 3.5. Positive Correlation of Notch3 and GSK3β Expression in Patients with Breast Cancer

We analyzed the protein levels of Notch3 and GSK3β in the pathological parameters of 68 human breast cancer samples. Forty-three of the 68 specimens (64.8%) were found to be positive for Notch3 expression. Notch3 negative and positive specimens are shown in Figure 5A,B, respectively. Positive GSK3β expression was found in 40 of the 68 samples (58.8%). Glycogen synthase kinase-3-beta negative and positive samples are shown in Figure 5C,D, respectively. Immunohistochemistry of Notch3 and GSK3β in breast cancer specimens revealed that Notch3 and GSK3β expression are positively associated (*r* = 0.416, *p* = 0.001; Table 1). Meanwhile, the expression abundance of Notch3 and GSK3β was associated with different breast cancer subtypes, and significantly increased in luminal breast cancer type compared with other subtypes (*p* = 0.037; Table 2). In general, these data demonstrated that Notch3 correlated in a positive manner with elevated GSK3β expression.

### 3.6. High Expression of Both Notch3 and GSK3β mRNA Predicts an Improved Prognosis in Breast Cancer Patients

We first examined the prognostic effect of the expression of Notch3 and GSK3β on www.kmplot.com (accessed on 1 April 2022). High expression of Notch3 (*p* = 4.1 × 10^−7^) and GSK3β (*p* = 3.8 × 10^−5^) mRNAs correlated with an improved RFS for all patients with breast cancer (Figure 6A,B). Restricting the analysis to intrinsic subtypes of breast cancer, we found that high expression of Notch3 correlated with a greater RFS for luminal A, luminal B, and human epidermal growth factor receptor 2 (Her2) subtypes but not the basal-like subtype compared to those with low expression of Notch3 (Figure 6D,G,J,M). In addition, patients with luminal A, luminal B, and Her2 subtypes showed a superior RFS when GSK3β was overexpressed but not the basal-like subtype (Figure 6E,H,N,K). Notably, high mRNA expression of both Notch3 and GSK3β predicted greater relapse-free survival in overall (*p* = 1.8 × 10^−5^) and luminal A breast cancer patients (*p* = 0.0082; Figure 6C,F), but not luminal B, basal-like or Her 2 subtypes (Figure 6I,L,O). In conclusion, Notch3 and GSK3β mRNA overexpression suggest a good prognosis for patients with breast cancer.

## 4. Discussion

The Notch receptor is a highly conserved type I transmembrane glycoprotein that is involved in cell differentiation, proliferation, and survival, which play important roles in various tumors [31]. Notch1–4 has highly similar structures and corresponding functional role in mammals, but has a distinct function in normal breast development and breast cancer [32]; it has been identified that inhibition of Notch1 expression can reverse the EMT process of breast cancer cells, thereby inhibiting cell migration and invasion [33]. Furthermore, a clinical study found that Notch1 correlated to a poor prognosis in patients with breast cancer [34]. Notch2 overexpression predicts better overall survival in women with breast cancer, which is completely different from Notch1 [35]. In addition, Notch2 was found could inhibit tumor growth, and continuous activation of N2ICD increases apoptosis in human breast cancer cells [36]. In contrast, Notch4 has been considered to act as a carcinogen in breast cancer. Iori et al. discovered that Notch4 inhibited the ability of proliferation and invasiveness in TNBC cells,as well as reduced tumor volume and tumourigenicity of mouse xenografts; moreover, TNBC had a higher frequency of nuclear translocation of Notch4 [37]. A recent study also showed the carcinogenic effect of Notch4. Zhou et al. revealed that Notch4 overexpression can promote the EMT of breast cancer cells and maintain the self-renewal properties of mesenchymal breast cancer stem cells [38], which is of concern and is associated with endocrine drug resistance in breast cancer.

Although the basic structure of Notch3 bears a resemblance to other Notch receptor members, many differences exist in the intracellular domain as well as a special shorter transcription activation domain. The classic and well-known functions of Notch3 are mainly involved in the regulation of proliferation, differentiation and apoptosis of vascular smooth muscle cells. The mutation of Notch3 can lead to dysfunction of vascular smooth muscle cell maturation or dysfunction of vascular artery differentiation [39]. In recent years, increasing evidence indicate its crucial role in the progression of cancer; it has been shown to have carcinogenic and tumor-suppressive effects in different cancers. Some studies suggested that it acts as an oncogene [40,41]. Nevertheless, several studies have indicated that Notch3 may play a role in inhibiting tumor development in breast cancer. For example, studies have indicated that Notch3 may play a considerable role in the formation of mammary epithelial cells during breast development [42]. Daniel et al. [43] revealed that Notch3 could label luminal progenitor cells and inhibit the proliferation of tumor cells. Cui et al. [14] found that Notch3 regulated cell senescence by regulating P21, thus inhibiting the occurrence and development of tumors. Similarly, Notch3 is associated with the inhibition of cell proliferation and apoptosis in HER2-negative breast cancer [44]. Meanwhile, it was suggested that Notch3 could upregulate Cyclin D1 and causes the accumulation of p27Kip which results in cell cycle arrest at the G0/G1 phase [45]. Gu et al. demonstrated that Fos-related antigen 1 (Fra1) activated the EMT process and was negatively regulated by Notch3 in MCF-7/ADM cells. Notch3-Fra1 signaling pathway mediated chemoresistance via the EMT [46]. Besides, our previous studies revealed that Erα inhibits EMT by suppressing Bmi1 [17], and Notch3 upregulates Erα [24]. Meanwhile, another study from our group showed that Notch3 can inhibit EMT in breast cancer epithelial cells by transactivating GATA3 [18] and Kibra [47]. Therefore, Notch3 may have a pivotal role in tumor suppression, especially in breast cancer EMT.

The loss of E-cadherin expression is regarded as an essential event in EMT; it leads to disruption of epithelial cell polarity induced EMT [48], thus maintaining the mesenchymal phenotype and enhancing the invasion and metastasis of cancer cells [38]. A variety of molecular mechanisms can repress E-cadherin expression via transcriptional inhibition and promotor methylation. Doble et al. demonstrated that GSK3β acts as an EMT suppressor of the zinc-finger transcriptional repressor, snail, which regulates epithelial marker E-cadherin [20]; these results indicated that both Notch3 and GSK3β inhibit EMT in breast cancer. Some studies suggested that there may be crosstalk between GSK3β and Notch. Foltz et al. found that GSK3β can physically bind and phosphorylate the intracellular domain of two Notch paralogs (Notch1 and Notch2). The activity of GSK3β protects the N1ICD from proteasome degradation [21]. Unlike Notch1, GSK3β phosphorylates the intracellular segment of Notch2, thereby inhibiting transcriptional activation of the Notch target gene [22]. Previous studies demonstrated that Notch1 does not bind to the promoter of GSK3β, yet Notch2 seems bind to it [10]; these observations have shown the crosstalks between GSK3β and Notch, both GSK3β and these two Notch paralogs colocalize in the nucleus in the majority of the Notch-IC-expressing cells; however, the interaction of Notch3 and GSK3β in EMT, a potential key step in breast cancer metastasis that contributes to breast cancer-related deaths, remains unclear.

In this study, our results revealed that GSK3β was highly expressed in the luminal-type breast cancer cell line MCF-7, but was expressed at low levels in the TNBC cell line MDA-MB-231; this is consistent with our previous findings that the expression pattern of Notch3 is similar to GSK3β in these cell lines. In addition, our previous studies found that the CSL-binding element, GGGAA, participated in regulating the GATA3 [18] and ER [24]. In this study, we demonstrated that Notch3 bound to the *GSK3β* promoter, which contains CSL-binding elements. Here, we found that the luciferase reporter activity of GSK3β decreased in the form of a concentration gradient after Notch3 knockdown in MCF-7 cells. In contrast, GSK3β-mediated luciferase activity increased after N3ICD over-expression in MDA-MB-231 cells. Our study on its molecular mechanism revealed that inhibition of Notch3 downregulated GSK3β mRNA and protein levels in MCF-7 cells significantly. Conversely, N3ICD overexpression upregulated that of GSK3β in MDA-MB-231 cells as illustrated in the Graphic Abstract.

In summary, our research findings indicated that Notch3 represses the processes of EMT in breast cancer, which is consistent with our previous reports. Remarkably, this finding demonstrates for the first time that Notch3 may inhibit the EMT process in breast cancer cells through transcriptionally upregulating GSK3β. From data obtained from clinical tissue, Notch3 expression was significantly consistent with GSK3β expression in breast cancer tissue samples. In a prognostic analysis, high expression of mRNA of both Notch3 and GSK3β was related to better RFS in all patients with breast cancer studied, which implies that Notch3 and GSK3β are beneficial biomarkers in this disease. Our results highlight a novel mechanism for exploring how Notch3 regulates EMT as well as the crosstalk between Notch and Wnt signaling pathways; this may have important implications for identifying new biomarkers for the prognosis of and as therapeutic targets in breast cancer.

## Figures and Tables

**Figure 1 cells-11-02872-f001:**
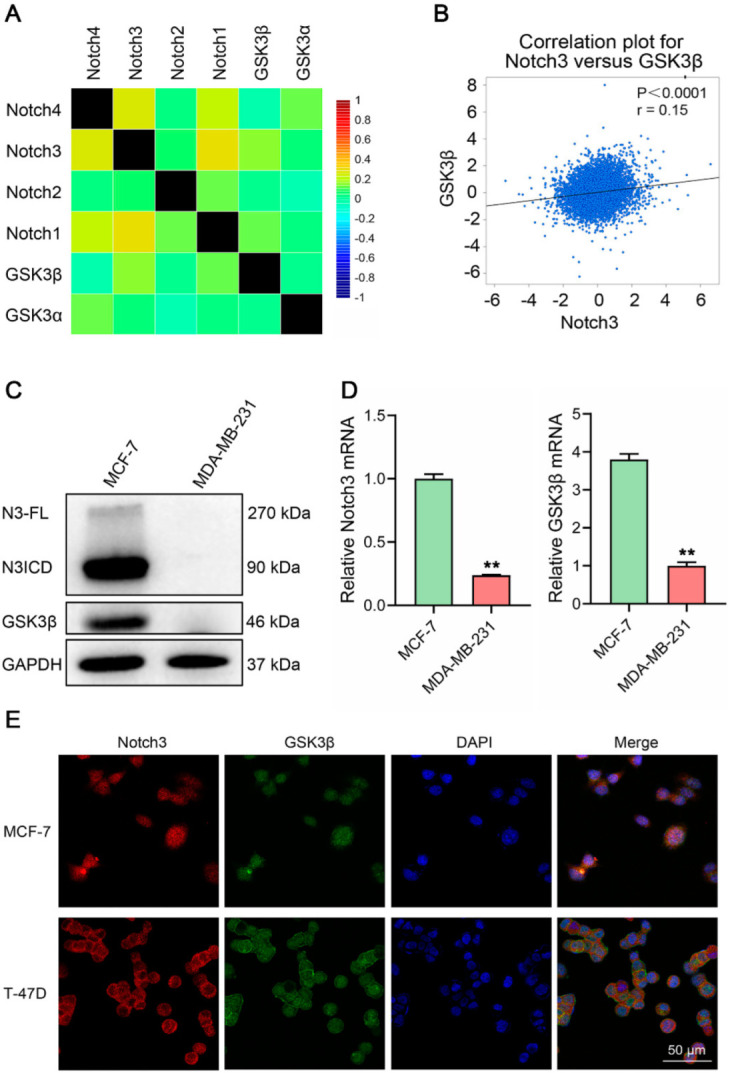
Notch3 is expressed in the luminal subtype and modulates GSK3β expression in breast cancer cell lines. (**A**): Heat map representing the correlation of DNA microarrays of GSK3β and Notch1–4, which was obtained from Breast Cancer Gene−Expression Miner v4.7. (**B**): A significant positive correlation existed between Notch3 and GSK3β. The correlation coefficient was *r* = 0.15, *p* < 0.0001. (**C**): Notch3 and GSK3β expression in distinct subtypes of breast cancer cell lines detected by western blotting. (**D**): Expression of Notch3 and GSK3β mRNA analyzed by RT–PCR in breast cancer cell lines. (**E**): Immunofluorescence staining of GSK3β and Notch3 in MCF-7 and T-47D cells. Nuclei were counterstained with DAPI. The scale bar represents 50 μm. ** *p* < 0.01.

**Figure 2 cells-11-02872-f002:**
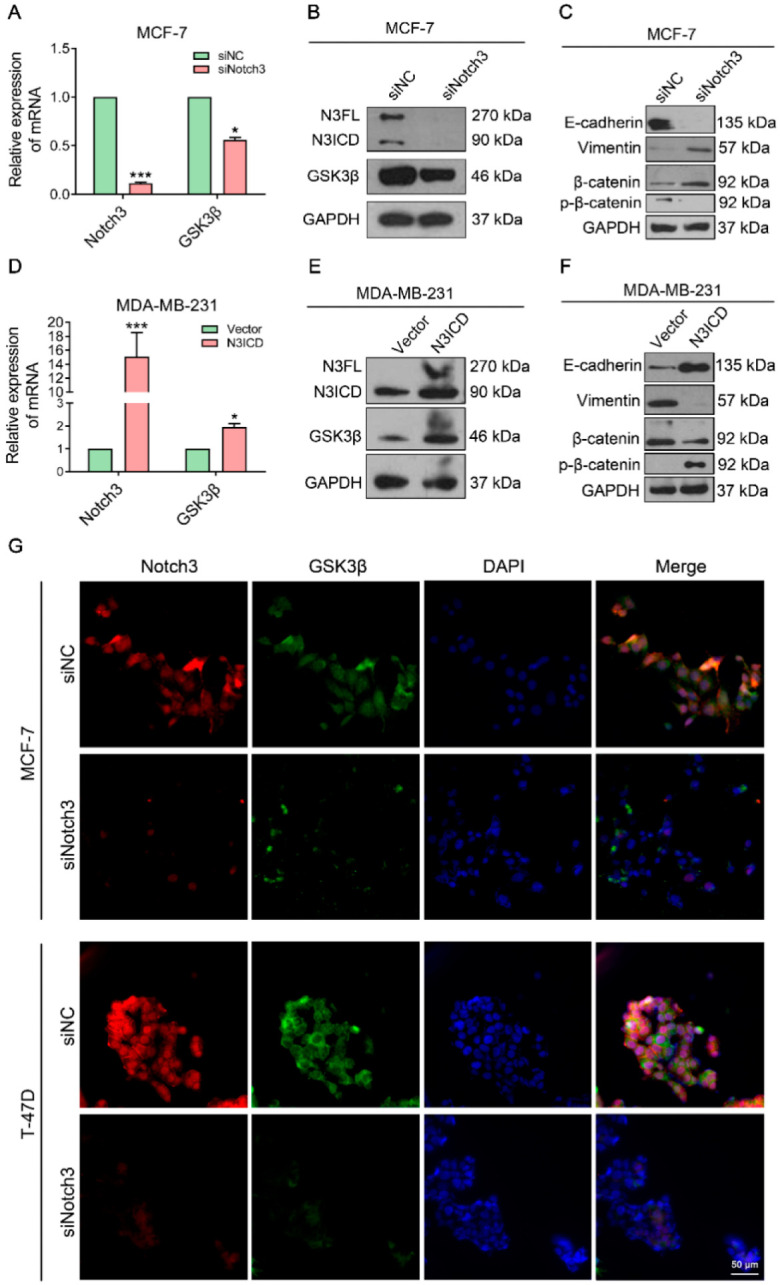
Ectopic Notch3 induces GSK3β expression and inhibits epithelial–mesenchymal transition (**A**): MCF-7 cells were transiently transfected with small interfering (si) RNA of Notch3 or vehicle (control). After 48 h, quantitative reverse transcription (qRT)–PCR was used to assess the mRNA levels of Notch3 and GSK3β. (**B**,**C**): Protein levels of Notch3, E-cadherin, and vimentin in MCF-7 cells transfected with siRNA-negative control (NC) or siRNA-Notch3 after 48 h were measured by western blotting. (**D**): MDA-MB-231cells were transiently transfected with pCLE-Notch 3 intracellular domain (N3ICD) or pCLE vehicle (control). After 48 h, qRT–PCR was used to assess the mRNA levels of Notch3 and GSK3β. (**E**,**F**): Protein levels of Notch3, E-cadherin, and vimentin in MDA-MB-231 cells transfected with pCLE-N3ICD or vehicle (control) after 48 h were measured by western blotting. (**G**): Immunofluorescence staining of GSK3β and Notch3 in MCF-7 and T-47D cells treated with control siRNA or siNotch3. Nuclei were counterstained with DAPI. The scale bar represents 50 μm.* *p* < 0.05, *** *p* < 0.001.

**Figure 3 cells-11-02872-f003:**
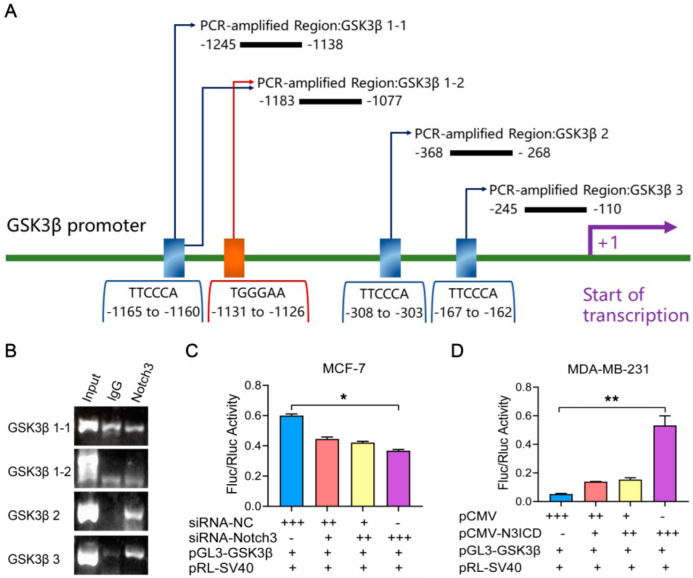
Notch3 transactivates GSK3β by directly binding to the *GSK3β* promoter. (**A**): A schema of the four CSL-binding element-containing primers (GSK3β 1−1, GSK3β 2, and GSK3β 3 regions containing respective single CSL-binding elements; region GSK3β 1−2 containing two CSL-binding elements) used for chromatin immunoprecipitation (ChIP) assays. (**B**): Products of ChIP were amplified by PCR and analyzed by 2% agarose gel electrophoresis. A specific band was seen in the GSK3β 2 region; (**C**): MCF-7 cells co-transfected with gradient concentrations of siRNA-Notch3 (“+” 10 pmol, “++” 20 pmol, “+++” 40 pmol, “−” 0 pmol) or the same concentrations of siRNA-negative control (NC) (“+” 20 pmol, “++” 30 pmol,“+++” 40 pmol,“−” 0 pmol), 10 ng pRL-SV40, and 200 ng pGL3-GSK3β-enhancer. (**D**): MDA-MB-231 cells co-transfected with gradient concentrations of pCMV- N3ICD; “+” 200 ng, “++” 400 ng, “+++” 800 ng, “−” 0 ng) or the same concentrations of pCMV (“+” 400 ng, “++” 600 ng, “+++” 800 ng, “−” 0 ng), 10 ng pRL-SV40, and 200 ng pGL3-GSK3β-enhancer. * *p* < 0.05, ** *p* < 0.01.

**Figure 4 cells-11-02872-f004:**
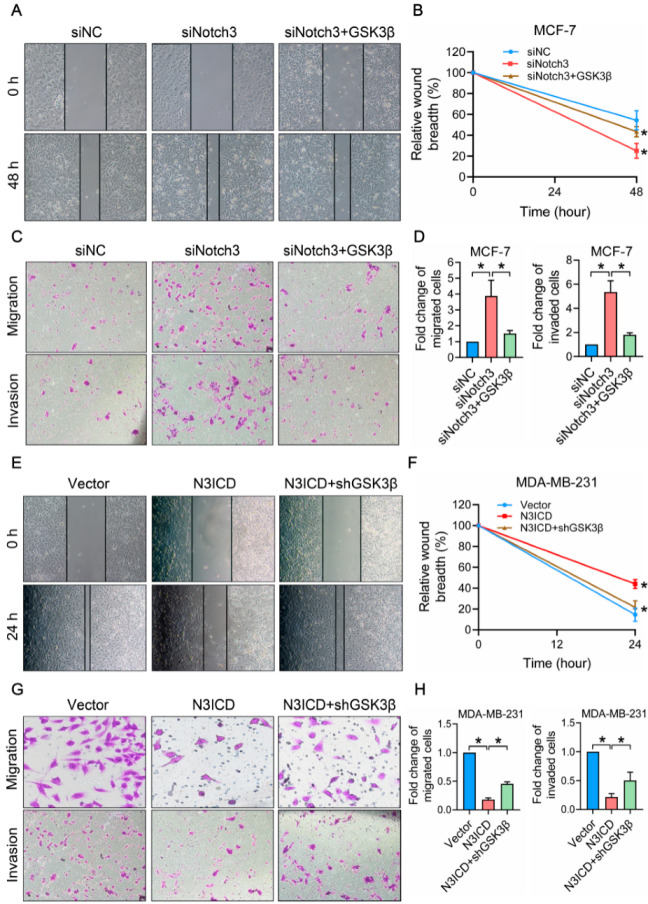
Restoration of GSK3β expression reverses Notch3-mediated suppression of migration and invasion in breast cancer cells. (**A**): Notch3 inhibition induced MCF-7 cell growth in vitro; this effect was attenuated by GSK3β overexpression. (**B**): Quantitative analysis of the wound healing rate of cells in each treatment group MCF-7siRNA-NC, MCF-7siRNA-Notch3 and MCF-7siRNA-Notch3 + GSK3β. (**C**,**D**): MCF-7siRNA-Notch3, MCF-7siRNA-NC, and MCF-7siRNA-Notch3 + pCMV3-GSK3β-GFP-Spark cells were subjected to transwell migration and invasion analysis. The mean ± SD of migrated cells of three independent experiments is shown in the panel. (**E**): N3ICD overexpression inhibited MDA-MB-231 cell growth in vitro; this effect was attenuated by GSK3β knockdown. (**F**): Quantitative analysis of the wound healing rate of cells in each treatment group above. (**G**,**H**): MDA-MB-231pCMV, MDA-MB-231pCMV-N3ICD, and MDA-MB-231pCMV-N3ICD + psi-U6.1/eGFP/shRNA-GSK3β cells were subjected to transwell migration and invasion analysis. The mean ± SD of migrated cells of three independent experiments is shown in the panel. Magnification 400×. * *p* < 0.05. GFP, green fluorescent protein; NC, negative control; siRNA, small interfering RNA.

**Figure 5 cells-11-02872-f005:**
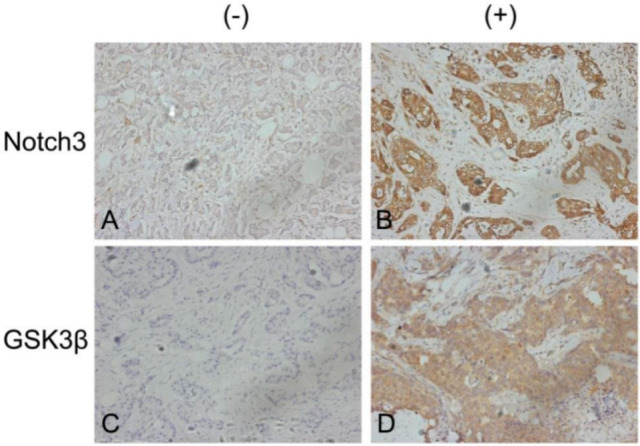
The correlation between Notch3 and GSK3β expression in human breast cancer specimens. (**A**): Representative image of Notch3 negative (−) staining cells in human breast cancer tissue. (**B**): Representative image of Notch3 positive (+) staining cells. (**C**): Representative image of GSK3β negative (−) staining cells. (**D**): Representative image of GSK3β positive (+) staining cells. Magnification 200×.

**Figure 6 cells-11-02872-f006:**
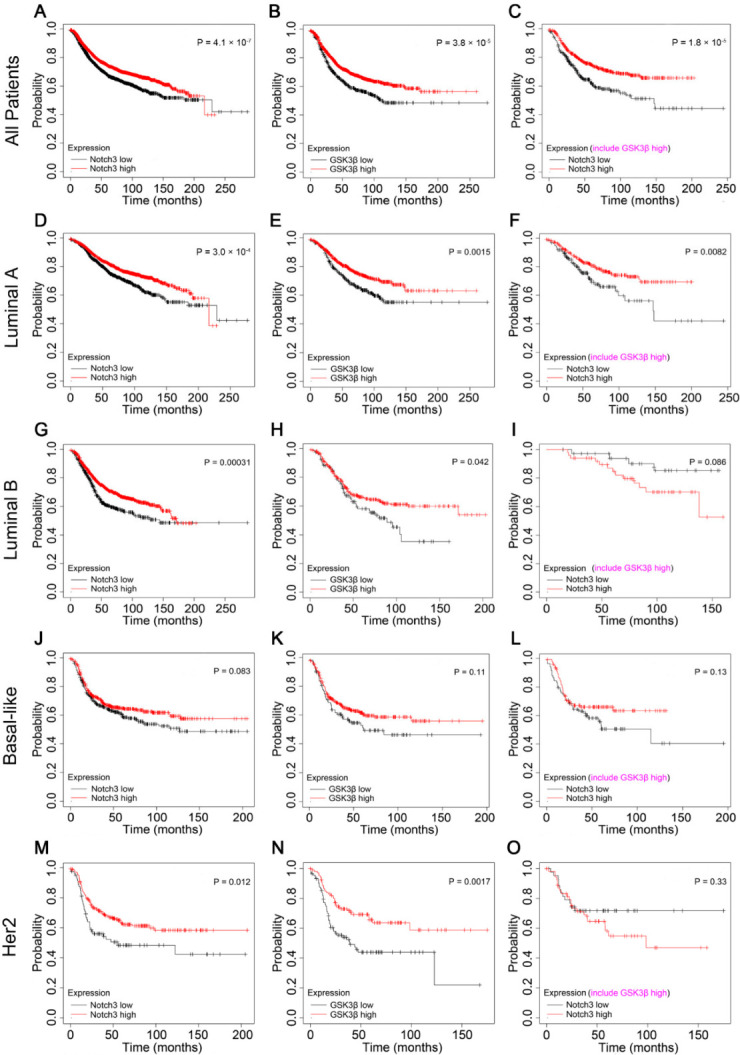
Analysis of Notch3 and GSK3β expression of RFS in patients with breast cancer. (**A**,**D**,**G**,**M**): High expression of Notch3 resulted in better recurrence-free survival (RFS) among different breast cancer subtypes but not the basal-like subtype (**J**). (**B**,**E**,**H**,**N**): High expression of GSK3β resulted in a better RFS among different breast cancer subtypes but not the basal-like subtype (**K**). A superior RFS was observed for breast cancer patients expressing both high Notch3 and GSK3β levels in all patients, luminal A subtype (**C**,**F**) but not luminal B, basal-like, or Her 2 subtypes (**I**,**L**,**O**).

**Table 1 cells-11-02872-t001:** Association of Notch3 and GSK3β expression in human breast cancer.

Notch3	GSK3β		χ2	*r*	*p*
	+	−			
+	32	11	11.744	0.416	0.001
−	8	17			

**Table 2 cells-11-02872-t002:** Correlation of Notch3 or GSK3β with clinicopathological features in 68 patients with breast cancer.

Clinicopathologic Features	Notch3-GSK3β- (*n* = 16)	Notch3 + GSK3β- (*n* = 9)	Notch3-GSK3β+ (*n* = 9)	Notch3 + GSK3β+ (*n* = 34)	*p*
Age at diagnosis					0.577
<50	7 (30.4%)	3 (13%)	4 (17.4%)	9 (39.1%)	
≥50	9 (20%)	6 (13.3%)	5 (11.1%)	25 (55.6%)	
Tumor stage (T)					0.679
T1	6 (27.3%)	3 (13.6%)	1 (4.5%)	12 (54.5%)	
T2	9 (24.3%)	5 (13.5%)	7 (18.9%)	16 (43.2%)	
T3	1 (25%)	1 (25%)	0 (0%)	2 (50%)	
T4	0 (0%)	0 (0%)	1 (20%)	4 (80%)	
LN stage (N)					0.735
N0	5 (20%)	3 (12%)	4 (16%)	13 (52%)	
N1	7 (26.9%)	4 (15.4%)	5 (15.4%)	11 (42.3%)	
N2	1 (9.1%)	2 (18.2%)	1 (9.1%)	7 (63.6%)	
N3	3 (50%)	0 (0%)	0 (0%)	3 (50%)	
Breast cancer subtypes					0.037
Luminal A	8 (32%)	7 (28%)	4 (16%)	6 (24%)	
Luminal B	4 (15.4%)	2 (7.7%)	3 (11.5%)	17 (65.4%)	
HER2	2 (20%)	0 (0%)	0 (0%)	8 (80%)	
TNBC	2 (28.6%)	0 (0%)	2 (28.6%)	3 (42.9%)	

## Data Availability

The datasets of Notch3 and GSK3β of Figure 6 (Affy ID: 1/Notch3: 203238_s_at; 2/GSK3β: 242336_at; 3/ Notch3, include GSKβ high: 203238_s_at and 242336_at; Survival: RFS. http://kmplot.com/analysis/index.php?p=service) and Appendix A (Affy ID: 1/Notch3: 203238_s_at; 2/GSK3β: 242336_at; 3/ Notch3, include GSKβ high: 203238_s_at and 242336_at; Survival: OS. http://kmplot.com/analysis/index.php?p=service) can be viewed on the website described above.

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
