# Peer review of "Notch3 Transactivates Glycogen Synthase Kinase-3-Beta and Inhibits Epithelial-to-Mesenchymal Transition in Breast Cancer Cells"

_cells, 2022, doi:10.3390/cells11182872_

Round 1

Reviewer 1 Report

The manuscript by Chen a collaborators entitled “Notch3 Inhibits Epithelial-to-Mesenchymal Transition by Transactivating Glycogen synthase kinase-3-beta in Breast Cancer” describes the potential crosstalk between the Notch3 signalling pathway and GSK3b.

The MS has some interesting data, for example the binding of N3ICD to the promoter of GSK3b, and some of the in vitro and clinical data.

Yet, the work includes claims that are not supported by the data, or that are incorrect.

Authors make a confusing statement about the “over-expression” of Notch3 (or GSK3b) as over-expressing the full-length Notch3 would not necessarily mean more activity – it would still need to be activated by one of the ligands, and even depending of the ligand the response might differ. I’m assuming they are over expressing the N3ICD. This needs to be clearly stated.

As well as direct interaction vs secondary effects.

For example:

Suppression of activated Notch3 (N3ICD) by siRNA”

siRNAs would downregulate the expression (mRNA) of Notch3, not the activity. In fact, there is little on this work on the activity of this receptor.

Same here:

Conversely, Notch3 was ectopically overexpressed in MDA-MB-231 cells by stably co-transfecting PCLE/N3ICD; this revealed that Notch3 overexpression up-regulated E-cadherin

What is PCLE? There is no direct evidence that N3ICD up-regulated CDH1. Instead it results in up-regulation.

This one is even more confusing:

This noteworthy finding demonstrated that ectopic overexpression of Notch3 decrease total β-catenin but increased β-catenin phosphorylation (Thr41/Ser37/Ser33), and endogenous silencing of Notch3 increased total β-catenin but decreased β-catenin phosphorylation (Thr41/Ser37/Ser33)

I’m assuming it is overexpression of N3ICD. Then the “decrease” of b-catenin is indirect somehow or it (indirectly?) results in more b-cat phosphorylation. While silencing N3 reversed this trend. Yet, there is no direct interaction between N3ICD and b-catenin, or is there?

This statement is incorrect:

As shown in Fig. 2G, Notch3 and GSK3β co-localized in the luminal breast cancer cell lines, MCF-7 and T-47D.

Co-localisation cannot be defined by low resolution microscopy. These proteins could be in the same cellular compartment, but from that that they are interacting is too much of a call. There is also no proper controls for this, for example KO/KD cells and/or overexpression. The specificity of the antibodies is questionable. And in practice there is relatively low “co-localisation” anyway. This claim cannot be used.

Conversely, the effect was reversed by ectopic GSK3β expression as the width of the wound was restored to 43.5%, indicating that the increased migration caused by the silencing of Notch3 was mediated by GSK3β (Fig. 4A and B)

One can talk about compensation not direct effects. That overexpressing GSK3b compensates partly for the effects of siRNA-N3, does not mean they “mediate” each other but they can compensate some of the effects. These 2 pathways are parallel and crosstalk to each other. This needs to be clear.

Discussion:

Notch1–4 has a highly similar structure

Notch1-4 have highly similar structures

The differences between Notch1 and Notch4 to Notch3 in breast cancer should be better discussed. Also the effect of kinases on the NICDs of these receptors, as phosphorylation changes their stabilities and activities – these are clearly different between N1ICD and N3ICD for example. Important, as activity is the key here. These issues should be better debated in the discussion as they matter to this work.

Also, while the role of Notch1 is better known as a tumour enhancer in breast cancer, the Notch receptors are also pleiotropic, and their prognosis and responses seem to vary a lot depending of the tissue. For example, according to protein atlas Notch3 is a prognostic marker in renal cancer (unfavourable) and endometrial cancer (unfavourable). The same applies to H&N cancers for example. Yet, the complexity of Notch signalling, where we are nto talking about expression but activity and the crosstalk to other pathways, makes the mRNA and even protein expression profiles

The same applies to GSK3b, which according to the protein atlas it is inverse correlated to survival(https://www.proteinatlas.org/ENSG00000082701-GSK3B/pathology/breast+cancer)

 More clinical data needs to be provided to support the conclusions as some of the other databases do not seem to support the claims. More immunohistochemistry analyses are needed too.

Author Response

Comment: The manuscript by Chen a collaborators entitled “Notch3 Inhibits Epithelial-to-Mesenchymal Transition by Transactivating Glycogen synthase kinase-3-beta in Breast Cancer” describes the potential crosstalk between the Notch3 signalling pathway and GSK3b.

 The MS has some interesting data, for example the binding of N3ICD to the promoter of GSK3b, and some of the in vitro and clinical data.

Yet, the work includes claims that are not supported by the data, or that are incorrect.

 Authors make a confusing statement about the “over-expression” of Notch3 (or GSK3b) as over-expressing the full-length Notch3 would not necessarily mean more activity – it would still need to be activated by one of the ligands, and even depending of the ligand the response might differ. I’m assuming they are over expressing the N3ICD. This needs to be clearly stated.

As well as direct interaction vs secondary effects.

For example:

“Suppression of activated Notch3 (N3ICD) by siRNA”

siRNAs would downregulate the expression (mRNA) of Notch3, not the activity. In fact, there is little on this work on the activity of this receptor.

Reply: Thank you for your comments. I agree with you that the full-length Notch3 does not necessarily mean more activity, and its activity sometimes depends on the level of ligand. As we described in the manuscript, Notch intracellular domain (NICD) is an active fragment of Notch receptor. Thus, in this study we overexpressed N3ICD to study the mechanism and function of active form of Notch3 in vivo and in vitro, rather than the full-length fragment of Notch3.

The statement “Suppression of activated Notch3 (N3ICD) by siRNA” is also confusing. We wanted to investigate whether inhibition of Notch3 by siRNA will affect the function of GSK3β. In fact, inhibition of Notch3 by siRNA, ultimately, resulting in N3ICD inhibition, significantly prevented transcriptional activation of GSK3β. These results were supported by N3ICD expression detected by Western Blots, and reporter assay with dual-luciferase vector.

We have revised “Suppression of activated Notch3 (N3ICD) by siRNA” in to “suppression/inhibition of Notch3 by siRNA” in the revised manuscript to avoid confusing.

Comment: Same here: Conversely, Notch3 was ectopically overexpressed in MDA-MB-231 cells by stably co-transfecting PCLE/N3ICD; this revealed that Notch3 overexpression up-regulated E-cadherin

What is PCLE? There is no direct evidence that N3ICD up-regulated CDH1. Instead it results in up-regulation.

Reply: Thank you for your comments. I am sorry for the confusing. Actually, pCLE is a control vector of pCLE-N3ICD plasmid. Accordingly, we have revised the sentence as ‘this revealed that Notch3 ICD overexpression resulted in E-cadherin up-regulation.’

Comment:This one is even more confusing:

This noteworthy finding demonstrated that ectopic overexpression of Notch3 decrease total β-catenin but increased β-catenin phosphorylation (Thr41/Ser37/Ser33), and endogenous silencing of Notch3 increased total β-catenin but decreased β-catenin phosphorylation (Thr41/Ser37/Ser33)

I’m assuming it is overexpression of N3ICD. Then the “decrease” of b-catenin is indirect somehow or it (indirectly?) results in more b-cat phosphorylation. While silencing N3 reversed this trend. Yet, there is no direct interaction between N3ICD and b-catenin, or is there?

Reply: Thank you for your comments. As we know, β-catenin, a key downstream protein of GSK3β in the classical Wnt signaling pathway, can be phosphorylated by GSK3β to form phosphorylated β-catenin, which is subsequently ubiquitinated and degraded, thus blocking the expression of Wnt signaling pathway (Cell.2017 Jun 1;169(6):985-999). We wanted to see the change of downstream proteins when GSK3 β was up-/down-regulated by Notch3.

In this study, we demonstrated that overexpression of N3ICD or inhibition of Notch3 by siRNA could up- and down-regulation of GSK3β expression. In the following studies, we found that the expression level of β-catenin was also altered. Therefore, we changed the sentence as “A noteworthy finding demonstrated that ectopic overexpression of N3ICD resulted in increase of β-catenin phosphorylation (Thr41/Ser37/Ser33), ultimately decreased total β-catenin, and conversely silencing of Notch3 by siRNA resulted in decreased β-catenin phosphorylation (Thr41/Ser37/Ser33), ultimately increased total β-catenin expression”. However, the relationship between Notch3 and β-catennin needs to be further explored in future studies.

Comment:This statement is incorrect:

As shown in Fig. 2G, Notch3 and GSK3β co-localized in the luminal breast cancer cell lines, MCF-7 and T-47D.

Co-localisation cannot be defined by low resolution microscopy. These proteins could be in the same cellular compartment, but from that that they are interacting is too much of a call. There is also no proper controls for this, for example KO/KD cells and/or overexpression. The specificity of the antibodies is questionable. And in practice there is relatively low “co-localisation” anyway. This claim cannot be used.  

Reply: Thank you for your comments. The definition and description of "Co-localization" is not accurate enough. We disrupted the expression of Notch3 in MCF7/T47D with siRNA , and revealed that the fluorescence level of GSK3β decreased correspondingly, consistent with the changes of protein and mRNA levels (shown in revised figure 2), thus further confirming the co-expression relationship between Notch3 and GSK3β. Overall, it is more appropriate to change it into “Co-expression” as your advice, which has been revised and marked yellow.

Comment: Conversely, the effect was reversed by ectopic GSK3β expression as the width of the wound was restored to 43.5%, indicating that the increased migration caused by the silencing of Notch3 was mediated by GSK3β (Fig. 4A and B)

One can talk about compensation not direct effects. That overexpressing GSK3b compensates partly for the effects of siRNA-N3, does not mean they “mediate” each other but they can compensate some of the effects. These 2 pathways are parallel and crosstalk to each other. This needs to be clear.

Reply: Thank you for your comments. This statement might not be accurate. Accordingly, we have revised the sentence as"Conversely, the effect was reversed by ectopic GSK3β expression as the width of the wound was restored to 43.5%, indicating that the increased migration caused by the silencing of Notch3 was partly compensated by GSK3β (Fig. 4A and B).”

Comment: Discussion: Notch1–4 has a highly similar structure

The differences between Notch1 and Notch4 to Notch3 in breast cancer should be better discussed. Also the effect of kinases on the NICDs of these receptors, as phosphorylation changes their stabilities and activities – these are clearly different between N1ICD and N3ICD for example. Important, as activity is the key here. These issues should be better debated in the discussion as they matter to this work.

Reply: Thank you for your comments. I agree with you that the differences between Notch1 and Notch4 to Notch3 in breast cancer should be intensely discussed. We have added relevant discussion content in the section of discussion and all marked yellow. For example, we have added discussions about the different roles that Notch1-4 play in breast cancer. Especially thecontroversial role of Notch3.

Comment: Also, while the role of Notch1 is better known as a tumour enhancer in breast cancer, the Notch receptors are also pleiotropic, and their prognosis and responses seem to vary a lot depending of the tissue. For example, according to protein atlas Notch3 is a prognostic marker in renal cancer (unfavourable) and endometrial cancer (unfavourable). The same applies to H&N cancers for example. Yet, the complexity of Notch signalling, where we are nto talking about expression but activity and the crosstalk to other pathways, makes the mRNA and even protein expression profiles

The same applies to GSK3b, which according to the protein atlas it is inverse correlated to survival (https://www.proteinatlas.org/ENSG00000082701-GSK3B/pathology/breast+cancer)

Reply: Thank you for your comments. I agree with you that the role of Notch3 in different cancers is still controversial. For example, some studies suggested that it acts as an oncogene, such as renal cancer and endometrial cancer. Nevertheless, several studies have indicated that Notch3 may play a role in inhibiting tumor development in breast cancer. For example, Notch3 is associated with the inhibition of cell proliferation and apoptosis in HER2-negative breast cancer (J Cell Biol. 2013 Oct 14;203(1):47-56.). Similarly, Notch3 was suggested could upregulate Cyclin D1 and causes the accumulation of p27Kip that results in cell cycle arrest at the G0/G1 phase (Cell Cycle. 2016;15(3):432-40.). Gu et al. clarified that increased fos-related antigen 1 (Fra1) activated the EMT process. Fra1 was negatively regulated by Notch3 and was highly expressed in MCF-7/ADM cells. Notch3-Fra1 signaling pathway mediates chemoresistance via the EMT (Tumour Biol. 2016 Oct 14.). Besides, our previous studies revealed that Notch3 may have a pivotal role in tumor suppression via mediating ERα、GATA-3、Kibra, especially in breast cancer EMT. According to protein atlas Notch3 is a prognostic marker in breast cancer, Notch3 high expression is associated with better survival outcomes (https://www.proteinatlas.org/ENSG00000074181-NOTCH3/pathology/breast+cancer). Thus, it seems to us Notch3 serves as a tumor suppressor.

We recognize that HPA database is a good tool for protein atlas analysis that can provide a lot of useful information.  However, this study mainly elaborated the synergistic relationship between Notch3 and GSK3β in breast cancer. HPA does not meet the above requirements which just analyze a single gene individually. Therefore, Kmplot database was used for survival analysis in this study. In addition, GSK3β is a key kinase in many malignancies, and its role in different cancers is controversial. A great deal of studies have found that it may promote the progression of colon, ovarian and prostate cancers (Cell Cycle. 2011 Jun 1;10(11):1717; Cell Res.2006 Jul;16(7):671-7; Cancer Lett.2012 Sep 28;322(2):177-84;), but inhibit the development of cancers such as breast, lung and skin squamous cell carcinoma (Cancer Res.2005 Jul 1;65(13):5792-801; Cancer Res.2007 May 15;67(10):4564-71; Anticancer Res. Sep-Oct 2007;27(5B):3561-9; Cancer Res.2007 Aug 15;67(16):7756-64). This may be related to the selectivity of GSK3β in different tumor types. Nevertheless, most studies support that the gene inhibits breast cancer progression, similar to our findings.

Comment: More clinical data needs to be provided to support the conclusions as some of the other databases do not seem to support the claims. More immunohistochemistry analyses are needed too.

Reply: Thanks for your comment. We provided additional clinical data to the association of Notch3 with GSK3β expression in the Table 2. The results from more databank further confirmed that Notch3 and GSK3β expression were associated with each other in different subtypes of breast cancer, and furthermore, the expression levels of Notch3 and GSK3β were significantly increased in luminal breast cancer.

Reviewer 2 Report

The manuscript called "Notch3 Inhibits Epithelial-to-Mesenchymal Transition by Transactivating Glycogen synthase kinase-3-beta in Breast Cancer" is well structured and can be published in this format in Cells from MDPI.

Author Response

The manuscript called "Notch3 Inhibits Epithelial-to-Mesenchymal Transition by Transactivating Glycogen synthase kinase-3-beta in Breast Cancer" is well structured and can be published in this format in Cells from MDPI.

Reply: Thank you for your comments. 

Reviewer 3 Report

For following comments, the authors may add response in Introduction and Discussion sections.

Since the expression level of Notch3 and GSK3β in MDA-MB-231 cell lines is low or even nearly absent, what is the rationale and significance to include triple negative cells in the transfection and inhibition assays?

Do the Notch3 and GSK3β play the same role in luminal cell type v.s. triple negative cell types in the EMT process?

Should Fig 7 be specified for luminal type breast cancer?  Or A diffenent Notch3ICD/ GSK3β axis model for triple negative cell type? 

Author Response

Comment: For following comments, the authors may add response in Introduction and Discussion sections.

Since the expression level of Notch3 and GSK3β in MDA-MB-231 cell lines is low or even nearly absent, what is the rationale and significance to include triple negative cells in the transfection and inhibition assays?

Reply: Thank you for your comments. We conducted the GSK3β inhibition assay in triple negative breast cancer cell with N3ICD overexpression and a rescue study in vitro to explore whether GSK3β could partially reverse the decline in cell migration and invasion caused by overexpression of Notch3.

Comment: Do the Notch3 and GSK3β play the same role in luminal cell type v.s. triple negative cell types in the EMT process?

Reply: In this study, we demonstrate for the first time that Notch3 and GSK3β expression trends are consistent in both luminal and triple-negative breast cancers; Mechanically, Notch3 inhibits the transformation of EMT and MET in luminal subtype cells and triple negative breast cancer by up-regulating GSK3β transcription. For example, the expression of epithelial marker E-CAD is increased, and the ability of cell migration and invasion is reduced.

Comment: Should Fig 7 be specified for luminal type breast cancer?  Or A diffenent Notch3ICD/ GSK3β axis model for triple negative cell type? 

Reply: Thank you for your comments. We think that Notch3ICD/ GSK3β axis model works for both luminal and triple negative cell type. The Fig 7 is for both.

Round 2

Reviewer 1 Report

I general the authors have amended most of my queries.

I do have a couple of additional issues:

I did some digging into Notch binding to promoter regions. Most of the published data is of Notch1. Where there is no binding to GSK3B promoter.

Sci Signal. 2017 May 2; 10(477): eaag1598.  doi: 10.1126/scisignal.aag1598

https://www.ncbi.nlm.nih.gov/pmc/articles/PMC5931361/

PNAS (2011) 108 (36) 14908-14913

https://doi.org/10.1073/pnas.11090231

https://www.pnas.org/doi/full/10.1073/pnas.1109023108

https://journals.plos.org/plosone/article?id=10.1371/journal.pone.0020022

https://www.ncbi.nlm.nih.gov/pmc/articles/PMC5264041/

No Notch1 binding to GSK3B, yet Notch2 seems to bind to it.

These works should be discussed as they show previous background work on the CSL/Notch biding, and the differences between the Notch receptors.

As a not, this reviewer is not an author in any of the articles mentioned.

Minor issues:

There is something wrong with the title. Please check it.

What is this section?

5. Patents

Patients?

Author Response

Dear Reviewer,

Thank you for providing us a valuable opportunity to make minor revisions to our manuscript (cells-1765598) ‘Notch3 Transactivates Glycogen synthase kinase-3-beta and Inhibits Epithelial-to-Mesenchymal Transition in Breast Cancer’. We have carefully read the reviewers’ comments, and made point-to-point responses accordingly. All revisions in the Introduction and Discussion sections were marked in red. The title of this article was also changed. I hope this revised version will satisfy you and the readers of your journal. I am looking forward to receiving your reply soon and eventually publishing this article in Cells.

The point-to-point responses are as follows:

I general the authors have amended most of my queries.

I do have a couple of additional issues:

I did some digging into Notch binding to promoter regions. Most of the published data is of Notch1. Where there is no binding to GSK3B promoter.

Sci Signal. 2017 May 2; 10(477): eaag1598.  doi: 10.1126/scisignal.aag1598

https://www.ncbi.nlm.nih.gov/pmc/articles/PMC5931361/

PNAS (2011) 108 (36) 14908-14913

https://doi.org/10.1073/pnas.11090231

https://www.pnas.org/doi/full/10.1073/pnas.1109023108

https://journals.plos.org/plosone/article?id=10.1371/journal.pone.0020022

https://www.ncbi.nlm.nih.gov/pmc/articles/PMC5264041/

No Notch1 binding to GSK3B, yet Notch2 seems to bind to it.

 These works should be discussed as they show previous background work on the CSL/Notch biding, and the differences between the Notch receptors.

As a not, this reviewer is not an author in any of the articles mentioned.

R: Thank you for your comments. We have added the important background of the distinct Notch/ transcription factor CSL binding sites in the Introduction and Discussion sections, respectively, as suggested by you, and all marked in red.

Details are as follows:

In the Introduction part, lines 20 to 28 of the first paragraph, we have added “In the canonical Notch signalling pathway, ligand binding induces the release and nuclear translocation of Notch receptor intracellular domains (NICDs), which interact with the transcription factor CSL, leading to subsequent activation of canonical Notch target genes. Liu et al. confirmed the finding that Notch1, Notch2, and CSL usually bind to different target genes, suggesting that the signalling activity of most Notch family members is relatively independent (PMID: 28123599). Similarly, genome-wide identification of Notch transcriptional complex binding sequence pairing sites has shown that Notch family members have different binding sequence pairing sites (PMID: 28465412, PMID: 21737748).”

In the Discussion part, line 13 to 14 of the third paragraph, we have added“Previous studies demonstrated that Notch1 dose not bind to the promoter of GSK3β, yet Notch2 seems bind to it(PMID:28123599).”

Minor issues:

There is something wrong with the title. Please check it.

R: Thank you for your comments. We have revised the title.

What is this section?

  1. Patents

Patients?

 R: Sorry for the typo. We have revised it as Supplementary Materials.

Sincerely yours,

Guo-Jun Zhang, MD, PhD

Executive director and Professor,

Department and Laboratory: The Cancer Center and the Department of Breast and Thyroid Surgery, Xiang’an Hospital of Xiamen University, School of Medicine, Xiamen University,2000 East Xiang'an Road, Xiamen, China.

Telephone number: +86-592-2184298.

E-mail: gjzhang@xah.xmu.edu.cn.